

# Heuristic-based load balancing for identical virtual machines: a fair scheduling approach using probabilistic methods

Abdullah Bajahzar

Department of Computer Science and Information, College of Science at Zulfi, Majmaah University, Al Majmaah, Saudi Arabia

Corresponding author
Abdullah Bajahzar,
a.bajahzar@mu.edu.sa

## ABSTRACT

The work that is considered in this article is the difficult task of designing a good scheduling policy for assigning the many activities involved in making multiple products to the available infrastructure of virtual machines (VMs). The objective of this study is to achieve an even workload entre virtual machines that are in charge of running manufacturing tasks. The ultimate goal of the research is to come up with a complete scheduling of all virtual machine-based tasks, with special attention to optimisation of a balanced level of lifetime as well as reduced variance between various virtual machines, which participate in the production activities. The main goal of this experiment is to mitigate the disparities in the turnaround time of VMs by allowing maintenance, or task handover to be done in a well-attuned way for a more uniform operational smoothness. It requires the optimization of the lowest work cycle of a virtual machine, which is very important to sustain the effective capabilities over time. Balancing the operational fairness of the VMs is recognised as the best scheduling policy for dealing with the problem difficulties. In this work, six different heuristic-based approaches are presented as feasible methods to solve the problem, based on mathematical formulations to provide a range of approximate solutions to the issues analyzed. The proposed approach is characterized by a probabilistic and iterative approach aimed at reinforcing the reliability of the obtained results. The results confirm that the proposed approximate solutions are effective, based on strict tests over 1,250 instances, with fixed metrics used to facilitate a comparison with all the heuristic algorithms. The experiments demonstrate that the repetitive-probabilistic heuristic dominates the other proposed heuristics in 82.2% of all instances, resulting in an average gap of 0.11 and time consumption of 0.036 s. The second-best heuristic, repetitive-mixed probabilistic heuristic, obtains 59.0% in percentage terms, the average gap is 0.28, and the running time is 0.034 s.

## INTRODUCTION

In the context of an industrial enterprise, the overarching goal that drives its operations is the recording of profits. The pivotal element that significantly influences the realization of

this goal is the effective and strategic management of resources and processes within the organization. Conversely, the objective that guides the operations of various information technology companies, particularly those that focus on the intricate workings of virtual machines, is fundamentally anchored in the sophisticated manipulation and optimization of the programs that are executed within these virtual environments. Additionally, guaranteeing the extended lifespan and lowering the depreciation rates of virtual machines is an important factor that requires attention for sustaining operational effectiveness and financial prudence as time passes. The rate of amortization attributable to the virtual machines is intrinsically linked to the cumulative number of working hours that each individual virtual machine is engaged in productive tasks, illustrating a direct correlation between usage intensity and value retention (*Fati et al., 2020*). It follows that a lower accumulation of working hours directly corresponds to a reduced rate of depreciation, thereby enhancing the financial viability of the virtual machine assets. This research endeavor is specifically designed to explore and identify a just and equitable distribution of operational strains across the different virtual machines, with the ultimate aim of diminishing the performance disparities that exist among them, a disparity that is quantitatively assessed as the differential in completion times when comparing the least burdened virtual machine to each of its counterparts. Additionally, it is essential to recognize that throughout each operational cycle, a virtual machine necessitates a certain duration for its cooling period, which incurs significant costs, and the extent of this requirement is heavily influenced by both the execution time and the volume of programs that have been processed by the virtual machine in question (*Kinger, Kumar & Sharma, 2014*).

With cloud computing becoming increasingly popular, the virtualization technology plays a crucial role, by abstracting the physical resources into virtual machines (VMs) in order to offer flexibility, and run complex workloads at scale. Workload scheduling to VMs in the multi-workload systems, the assignment of workload to VMs is a key problem which can directly affect the system performance, fairness and user satisfaction. Conventional workload assignment schemes may raise large disparities of completion time among VMs and may affect the efficency of the system or even the service level agreement (SLAC) violation of some VMs. To overcome these challenges, this work hones in on fair allocation, which seeks to reduce the inequality of workload completion times while also keeping utilisation high. Simplistic heuristics are proposed as lightweight and evolving measures to deal with the inevitable uncertainty and changes in workload demands. These heuristics provide an empirical tradeoff between the computational cost and the goodness of the allocation, and are suitable for large-scale systems deployed in practice. The proposed scheme is a systematic method to attain fairness, minimizing the disparities among the finish times, and maintaining an equal usage of resources in virtualized systems.

The primary objective of this research is to systematically diminish the disparities between virtual machines, thereby facilitating the unification of various interventions into a singular, coherent timeframe that can be effectively managed (*Ashraf, Byholm & Porres, 2018*). In the context of a specified collection of tasks that are to be allocated across multiple virtual machines, it is essential to recognize that each individual task possesses its

own distinct running duration and is assigned to a predetermined virtual machine that does not change (*Murad et al., 2024*). In order to achieve an optimal balance between the operational lifetimes of the virtual machines and the attenuation of their performance while handling complex manufacturing processes (*Lv, Fan & Wang, 2021*), A sophisticated mathematical model has been developed to address the intricacies of the defined problem; furthermore, a series of advanced algorithms has been formulated to provide effective solutions to this multifaceted issue. The fundamental approach to resolving this intricate problem is predicated upon the principle of maximizing the minimum total completion time across all scheduled jobs, ensuring that efficiency is achieved in the overall processing time.

A well-rounded approach to resource allocation and scheduling is crucial for maximizing the least running time on virtual machines, aiming to enhance the task or machine that shows the lowest efficiency, thereby significantly elevating the overall system performance. In the realm of cloud computing, the process of task scheduling emerges as a pivotal component for the enhancement of makespan and resource utilization; this is paramount as demonstrated by the research conducted by *Raeisi-Varzaneh et al. (2024)*, who have put forth an advanced max–min algorithm designed to strike a balance between the execution time of tasks and the efficient utilization of resources, thereby exhibiting superior performance compared to traditional algorithms in terms of both makespan and waiting time, as referenced in their study. In a similar vein, the research conducted by *Konjaang, Maipan-Uku & Kubuga (2016)* has placed a particular emphasis on refining the max-min scheduling algorithm to boost efficiency within cloud environments, thereby tackling the pressing issue of minimizing makespan by prioritizing the execution of tasks that possess the maximum execution time, as articulated in their findings. Furthermore, the investigation carried out by *Rampersaud & Grosu (2015)* has addressed the complexities inherent in the multi-resource sharing-aware virtual machine maximization problem by proposing a greedy approximation algorithm that aims to maximize profit while simultaneously taking into account the various constraints imposed by resource availability, as noted in their research. Moreover, the work conducted by *Shi & Xu (2014)* on the Cloud Utility Maximization model further underscores the critical nature of optimal placement strategies for virtual machines, as a means to maximize the utility derived from cloud computing resources, utilizing a subgradient algorithm to effectively resolve the Lagrangian relaxation dual problem, as detailed in their publication. Collectively, these scholarly contributions illuminate the profound significance of sophisticated scheduling and resource allocation strategies in the quest to maximize the minimum running time on virtual machines, thereby ensuring not only efficient utilization of resources but also a marked enhancement in performance within cloud computing environments.

Resource distribution, a crucial aspect within the sphere of cloud computing frameworks designed for extensive data processing, acts as a core strategy that ensures optimal resource usage, achieves peak performance standards, and fosters economic efficiency in a fiercely competitive online environment. The swiftly increasing demand for scalable applications, in conjunction with the unprecedented influx of data produced from

a multitude of varied sources, necessitates the deployment of robust and advanced load-balancing mechanisms within cloud computing ecosystems to adeptly address these challenges. Such mechanisms are meticulously designed with the primary objective of evenly distributing workloads across multiple servers, thereby significantly enhancing overall system performance metrics in terms of resource utilization, throughput levels, and response times, as highlighted in previous studies (*Patel, Mehta & Solanki, 2021*). Traditional load-balancing methodologies encounter multifaceted challenges, including issues related to adaptability and unpredictability inherent within cloud architectures, which has subsequently spurred the exploration and development of meta-heuristic algorithms that draw inspiration from natural processes such as ant foraging behaviors and predator hunting strategies. These innovative algorithms possess a remarkable capability to swiftly identify superior solutions in scenarios where conventional load-balancing techniques may struggle or falter, as documented in recent research (*Fahim et al., 2018*). The intricate nature of load balancing is classified as NP-complete, which signifies that it encompasses an extensive search space filled with a multitude of potential solutions, thereby rendering it a particularly challenging problem to resolve optimally, as noted in academic literature (*Ghomi, Rahmani & Qader, 2017*). A variety of strategic approaches, including but not limited to virtual machine migration and the scheduling of Hadoop queues, have been proposed in the academic discourse to tackle these formidable challenges, with a concentrated focus on addressing the specific demands of applications while ensuring a high level of performance and availability, all while striving to minimize both costs and energy consumption (*Dey & Gunasekhar, 2019*). These strategic methodologies are of paramount importance for upholding service level agreements (SLAs) and averting any degradation in performance within cloud-based data centers, as emphasized in scholarly research (*Dey & Gunasekhar, 2019*). In summary, the ongoing and dynamic evolution of load-balancing algorithms, which encompasses both static and dynamic approaches, is of critical importance for not only achieving high levels of user satisfaction but also for ensuring the efficient allocation of resources within cloud computing environments, as detailed in existing studies (*Patel, Mehta & Solanki, 2021*).

Cloud computing architectures fundamentally depend on virtualization technologies, wherein tangible resources are transformed into VMs to facilitate flexibility and scalability for a myriad of workloads. Within these frameworks, the distribution of workloads to VMs constitutes a pivotal endeavor that exerts a direct influence on overall system performance, equity, and user contentment. Conventional workload distribution methodologies frequently yield substantial variances in completion durations among VMs, culminating in inefficiencies and possible infractions of service-level agreements. To mitigate these issues, this research emphasizes equitable allocation, which aspires to diminish discrepancies in workload completion times while concurrently preserving elevated system utilization. Probabilistic heuristics are proposed as streamlined and adaptive techniques to navigate the intrinsic uncertainties and dynamic characteristics of workload demands. These heuristics present a viable equilibrium between computational efficiency and allocation excellence, rendering them particularly appropriate for practical large-scale systems. The suggested methodology furnishes a systematic framework for attaining fairness, lessening

completion time discrepancies, and guaranteeing just resource utilization in virtualized environments.

The load balancing (LB) capability is fundamental in cloud computing, formulated to spread workloads evenly across different servers to enhance resource management and lessen server crowding. A range of algorithms and architectures have been introduced to elevate LB functions, blending traditional tactics with cutting-edge machine learning solutions. Vital tactics entail the flexible sharing of tasks between VMs to shorten response intervals and enhance the system's overall effectiveness (*Ray & Singhal, 2024*). The use of Machine learning frameworks—specifically artificial neural networks (ANNs), random forest classifiers, and long short-term memory networks—is increasingly seen in LB methodologies, enabling adaptive workload management informed by performance indicators like throughput and fault tolerance (*Muchori & Mwangi, 2022*). The efficacy of these algorithms is vital for sustaining high user satisfaction and resource utility in cloud environments, where demand can vary considerably.

In the same context, in *Aghdashi & Mirtaheri (2019)*, a comprehensive investigation was conducted that centered around a sophisticated two-level job assignment mechanism specifically designed to address the intricate challenges associated with load balancing within the realm of cloud computing environments. In *Jemmali (2019)*, an innovative solution was meticulously proposed that aimed to enhance revenue distribution by employing the strategic maximization of the minimum revenue, thereby ensuring a more equitable allocation of resources. The approximate solution articulated in a recent publication has been judiciously utilized as a foundational reference to delineate and explore potential approximate solutions pertinent to the specific problem domain. Another significant area of application for the maximization of the minimum problem can be observed within the context of the aircraft industry, where it plays a crucial role in optimizing various operational parameters. In *Jemmali et al. (2019)*, a detailed examination is presented regarding the derivation and formulation of several lower bounds that pertain to the maximization of the minimum lifespan of a turbine engine, thereby contributing valuable insights to the field.

The contributions of the article are:

- Problem formulation based on fairness: A pioneering scheduling framework is presented that emphasizes equity by optimizing the minimum completion time across virtual machines. This objective diverges from conventional makespan optimization and has not garnered significant attention in the existing body of research.
- Design of six heuristic algorithms: Six robust and scalable heuristic algorithms, including probabilistic, repetitive, mixed, and reversed variants, are developed to tackle the proposed fair scheduling issue. Each variant employs a distinct methodology for task organization or selection to thoroughly explore the solution landscape.
- Simulation-based evaluation on extensive benchmarks: Extensive experiments are conducted on a comprehensive dataset consisting of 2,250 instances (encompassing both small and large scales) to evaluate the effectiveness of the proposed heuristics in terms of fairness and computational efficiency.

- Analysis under homogeneous cloud environments: The model assumes a set of identical virtual machines, with the analysis focused on clarifying load balancing dynamics within this practical yet simplified framework.
- Foundations for future extensions: Although the present study is simulation-based, it lays a foundational framework that can be modified for heterogeneous and dynamic environments, which are identified as potential directions for future research.

The significance of the proposed research is not just theoretical scheduling and applicable to practical industrial systems and computational systems. More specifically, four major areas are identified where the proposed method has the potential to have practical applications and impact:

1. Application in digital manufacturing and smart factory scheduling. This equitable workload allocation approach is consistent with the strategic requirements of smart factory systems, where the equal sharing of work of resources and the optimized use of resources is crucial to maintain performance and energy efficiency. In such environments, scheduling strategies need to prevent the computational resources (*e.g.*, edge and cloud nodes) from being overloaded, and that directly impacts the goal of decreasing production delays and alleviating system bottlenecks.
2. Suitability for cloud-based production systems: The proposed model is designed for virtualized environments, and thus very applicable to cloud-native production systems where multiple manufacturing services or simulated instances are distributed among virtual machines. The new heuristics enable effective VM-to-task allocations, which help manufacturers endorse the best VMs and enhance cloud resource efficiency among scalable, on-demand platforms.
3. Foundation for integration with digital twins: The probabilistic and adaptive nature of the scheduling approach adopted in the sprint scheduling algorithms provides a flexible base for integration with digital twin frameworks, which require real-time data simulations equilibrium to be preserved among computational nodes. Fair assignment of resources improves the predictability and responsiveness of simulations and the quality of mimics in an industry facility.
4. Contribution to computational sustainability and load fairness: In industrial cloud and cyberphysical systems, these fairness guarantees is of paramount importance to avoid large disparity in loads across computational units. Optimising the worst case completion time is important for sustainable computing, and ensuring that workloads are fairly divided is vital for the long term health and reliability of a system.

The subsequent sections of this article are structured as follows. 'Related Works' examines pertinent literature and underscores current research pertaining to workload scheduling and heuristic methodologies within cloud computing environments. 'Problem Definition' discusses the articulated problem statement, weaving in system assumptions, the pertinent mathematical expressions, and the notation patterns consistently adopted in this inquiry. A thorough examination of the heuristic algorithms, including their specific

adaptations, is laid out in 'Scheduling Algorithms'. This fifth portion takes a closer look at the experimental framework, the methodologies for producing instances, the evaluation metrics, and the insights that surfaced from the simulations performed in the experiments. To conclude, 'Conclusion' encapsulates the document comprehensively and suggests various paths for subsequent investigations.

## RELATED WORKS

In the case of cloud computing load balancing algorithms also play a vital role due to the dynamic and heterogeneous VM environments. Techniques like Daemon-COA-MMT improve fault tolerance and resource utilization through predicting possible failures and migrating applications on efficient hosts, hence minimizing energy consumption and increasing the reliability of the system (*Jahanpour, Barati & Mehranzadeh, 2020*). Meanwhile, dispatching work to VMs according to their processing ability facilitates a fair allocation of workloads and the goal can be minimizing the time to completion and maximizing the utilization of resources, which is essential in the elastic and scalable cloud environment (*Kaur & Ghumman, 2018*). These developments in load balancing algorithms through quantum annealing and VM-based approaches are crucial to improve the performance and efficiency of HPC systems.

Max-min fair allocation is a fundamental and challenging notion that carefully seeks to maximize the minimum utility or allocation of resources among all the parties in an allocation problem. In the particular setting of indivisible goods, the overall max-min fair allocation problem, commonly known as the Santa Claus problem in the academic literature, seeks to find a fair distribution of resources among multiple players such that the maximin utility of all players is maximized. This is a complicated problem that is nicely reformulated into a machine covering problem, and then a $\frac{c}{1-\varepsilon}$-approximate solution that can be computed in polynomial time by *Ko et al. (2021)*. In the centralized resource systems, the max-min fair allocation is hard to be achieved due to the strategic behaviors, in which the nodes may misbehave their actual demand/request or create the fake nodes that leading to obtain more resources. While incentivization strategies seem to provide the allocation mechanism some resiliency against the miscoordination problem, it is still highly vulnerable to a node splitting approach founded on the fact that nodes can significantly grow their share of the resources by dividing into several imaginary nodes (prefixed as a spitting strategy), as presented in the articles of *Chen, Gu & Wang (2021, 2020)*.

In the context of software-defined radio access networks (SD-RANs), the principle of max-min fairness is strategically employed to elevate network performance by effectively decoupling the control and data planes, which results in a remarkable increase of up to fourfold in the minimum data rates when compared to conventional systems, according to the findings presented by *Mehmeti & Kellerer (2022)*. Furthermore, in scenarios involving multi-source transmission, the attainment of max-min fairness necessitates the execution of a complicated joint optimization process that encompasses both bandwidth allocation

and flow assignment, thereby adding layers of complexity to the challenge. A novel methodological approach utilizing linear programming has been innovatively developed to proficiently achieve the goal of global max-min fairness, which has demonstrated superior performance over traditional methods in terms of enhancing network throughput and reducing transfer completion times, as documented in the research by *Li et al. (2019)*. Collectively, these studies illuminate the wide-ranging applications and inherent challenges associated with max-min fair allocation across various distinct domains, underscoring its significance in contemporary resource management discussions.

The algorithmic framework that facilitates the balancing process, which is extensively implemented within the realm of high-performance computing, can be specifically referenced in the scholarly works cited in this particular context, notably those found in *Arunachaleswaran, Barman & Rathi (2019)*, *Li et al. (2019)*. In their comprehensive inquiry, the authors delve deeply into the intricate complexities associated with the design and implementation of mechanisms that are aimed at generating allocations characterized by a constrained level of envy, thereby addressing a significant challenge in the field.

The implementation of load balancing techniques is strategically applied to enhance the efficiency and longevity of drone battery usage during operations. In their research, the authors have meticulously developed innovative algorithms specifically designed to maximize the utility and performance of battery systems throughout the entirety of the drone's flight operations (*Jemmali et al., 2022a*). Furthermore, the concept of load balancing has also been effectively applied within the domain of smart parking systems, thereby improving their operational efficiency and user experience (*Sun et al., 2020*). Moreover, the challenges associated with load balancing are comprehensively addressed across various sectors, including aviation, healthcare, industrial applications, and cloud computing environments, as evidenced by a substantial body of literature (*Jemmali et al., 2019*, *2022b*; *Dornala et al., 2023*; *Jemmali, Otoom & al Fayez, 2020*; *Jemmali, Melhim & Alharbi, 2019*; *Alquhayz & Jemmali, 2021*; *Eljack et al., 2024*).

The regression analysis meticulously examined in the works cited as *Salhi & Jemmali (2018a*, *2018b)* can indeed be effectively utilized to ascertain the specific hard classes that have been generated within the confines of this article, which addresses the complex problem that has been the focus of study. Furthermore, it is worth noting that a variety of alternative scheduling model types can be incorporated into the proposed algorithms as indicated in the research conducted by *Jemmali, Alharbi & Melhim (2018)*, thereby enhancing the versatility and applicability of the methodologies discussed. Numerous scholarly works have approached the critical issue of load balancing by implementing various scheduling algorithms designed to optimize performance and efficiency in diverse contexts. In the realm of smart parking management, where the challenges posed are recognized as NP-hard, achieving approximate solutions is poised to represent a significant advancement in the field, as highlighted in the findings presented in *Jemmali (2022)*.

Table 1 presents a comprehensive overview of the methodologies employed, the contributions made, and the limitations identified within the reviewed literature, while

**Table 1 Risk assessment methods and contributions.**

| Ref. | Methods used | Contributions | Limitations |
|---|---|---|---|
| *Kashani & Mahdipour (2022)* | The article categorizes load balancing algorithms in fog computing into four classifications: approximate, exact, fundamental, and hybrid algorithms. | This carefully crafted manuscript scrutinizes load-balancing algorithms in the domain of fog computing, organizing them into four defined categories: approximate, exact, fundamental, and hybrid algorithms. This categorization supports a detailed comprehension of the various approaches applied in load balancing throughout fog networks. | The manuscript emphasizes the absence of a comprehensive investigation aimed at synthesizing the scholarly contributions concerning load balancing algorithms within fog networks, thereby underscoring a notable deficiency in the existing literature that warrants rectification. |
| *Gures et al. (2022)* | The manuscript explores an array of load balancing strategies, encompassing the handover (HO) mechanism, which facilitates the transition of user equipment (UEs) at the cell boundary to cells with lower traffic loads, thereby attaining a more equitable distribution of load and enhancing overall system capacity. Effectively guiding the flow of traffic across various types of cells in extremely crowded and diverse networks (HetNets) is crucial. | The manuscript presents an extensive review of advanced load balancing frameworks designed for ultra-dense heterogeneous networks (HetNets), with an emphasis on machine learning (ML) methodologies. It delineates the overarching challenge of load balancing, encompassing its objectives, operational capabilities, assessment standards, and a fundamental operational framework, thus acting as a reference for the development of economically viable and adaptable ML-driven solutions. | The implementation of machine learning (ML) algorithms in addressing load balancing issues encounters considerable obstacles, especially concerning the handover (HO) process, which entails the relocation of user equipment (UEs) at the cell interface to cells with lower loads. Although this strategy can enhance cell load distribution and augment overall system capacity, it also presents intricacies that require proficient management. |
| *Zhou et al. (2023)* | This article presents a detailed comparative investigation of several metaheuristic load balancing strategies specifically crafted for cloud computing, concentrating on performance measures such as makespan length, imbalance level, response delays, data center processing duration, flow duration, and resource use effectiveness. | The manuscript presents an extensive and methodical review of the latest metaheuristic load-balancing algorithms, delivering an operational insight into these methodologies and their implementation within cloud computing ecosystems. It evaluates a range of algorithms, their classification, essential characteristics, and the obstacles related to load balancing. | The manuscript underscores that load balancing within the realm of cloud computing is characterized as an "NP-hard" problem, signifying that it possesses an extensive solution space that complicates the efficient identification of the optimal solution. This intricacy necessitates additional time to ascertain optimal solutions, which may impede performance in real-time applications. |

| Ref. | Methods used | Contributions | Limitations |
|---|---|---|---|
| *Nazir et al. (2022)* | The manuscript presents a newly formulated approach for load balancing within cloud computing at the database tier, with a particular emphasis on database cloud services that are commonly utilized by organizations for application development and business operations. | The document presents an innovative approach for optimizing load distribution in cloud computing, particularly at the database tier, which is essential for organizations of varying scales that leverage database cloud services for application development and operational processes. | The article's load balancing framework presents advancements but exhibits significant limitations. It is mostly suited for like server ecosystems, potentially falling short in varied cloud contexts. The framework also lacks a thorough security management system, despite recognizing its importance for cloud services. Moreover, it does not comprehensively tackle larger cloud challenges such as storage costs or overall capacity management. Although intended for scalability, the analysis of its performance during extreme user growth scenarios is limited. In conclusion, the review is based on defined, restricted contexts, which might not sufficiently symbolize a range of actual cloud operations or assure broader significance. |
| *Kulkarni et al. (2022)* | The report looks into techniques for improving productivity and asset utilization within cloud computing, accentuating load governance, task organization, resource handling, service excellence, and workload administration. | This analysis reviews the current techniques for balancing loads within cloud computing, underlining their challenges to support the refinement of superior algorithms. | The study underscores the intricacies of securing ideal load balancing within cloud computing architectures, especially in the context of dodging overload and underload situations for virtual machines, thereby affecting the comprehensive reliability and operational performance. |
| *Shakeel & Alam (2022)* | The manuscript examines load balancing (LB) algorithms within cloud and fog computing frameworks, offering an exhaustive classification of LB algorithms alongside an in-depth analysis utilizing heuristic, meta-heuristic, and hybrid methodologies. | The manuscript delineates a comprehensive categorization of load balancing (LB) algorithms explicitly designed for cloud and fog computing environments, tackling the intricacies and obstacles linked to the management of numerous dynamic user requests and congested virtual machines (VMs). | The manuscript underscores that load balancing within cloud-fog ecosystems constitutes an NP-hard problem, signifying the intricate nature and computational difficulties associated with attaining optimal load allocation across virtual machines. |

elucidating the fundamental distinctions among the diverse studies and emphasizing these aspects in the proposed approach.

Adaptive workload management in cloud computing is essential in order to meet SLAs and improve resource utilization. HEPGA is flexible to optimize different objectives: minimizing the makespan and enhancing the resource utilization when scheduling scientific workflows by the parallel scheduling (*Mikram, El Kafhali & Saadi, 2024*) based on the combination of the ideas of the heterogeneous earliest finish time (HEFT), particle swarm optimization (PSO) and genetic algorithm (GA). *Ghandour, El Kafhali & Hanini (2024)* described an adaptive model of workload management in cloud computing that maintains SLA compliance and high resource utilization while saving CO2 and costs. It is developed based on queuing theory and incorporates dynamic VM placement and energy efficient scheduling to balance loads in different scenarios. Simulation and AWS-based validation suggest markedly reduced response times, resource utilization, and fault tolerance in contrast to static methods. Such a model is an efficient, scalable and more importantly practical solution for dynamic cloud workload management. The chaotic hybrid particle swarm optimization (CHPSO) algorithm, a chi-squared particle swarm optimization based task scheduling algorithm for cloud computing was developed in *Mikram & El Kafhali (2025)*. Taupe created a chi-squared distribution for task arrival and based on total million instructions per second (MIPS), it guaranteed strategically and adaptively allocating tasks on CPU virtual servers. Empirical results demonstrate that CHPSO outperforms existing approaches in terms of response time, makespan, energy efficiency, and resource utilization. In *Jain, Jain & Tyagi (2025)*, an enhanced dynamic virtual machine consolidation (DVMC) model named RLSK_US was presented to handle the energy–SLAV tradeoff in cloud datacenters. The model is comprised of four phases: Robust Logistic Regression for detecting overloaded host, SLA-based analysis for detecting underloaded host, VM selection based on Knapsack and VM placement based on Utilization-SLA. The evaluation with real workload traces demonstrated that RLSK_US achieved a significant improvement over the benchmarks in terms of service level agreement violation (SLAV) (77% reduction) and energy saving value (ESV) (83% reduction).

# PROBLEM DEFINITION

The section Problem Definition is divided into three subsections to ensure a thorough understanding of the proposed method. The first subsection clarifies the notations and symbols utilized in the formulation. The second subsection outlines the problem context, including objectives, constraints, and challenges in equitable workload allocation within virtualized environments. Lastly, the third subsection details the mathematical model that formally articulates the problem and lays the groundwork for the proposed solutions.

## Notation

Table 2 presents the principal notations employed in this article. It specifies parameters pertinent to programs and virtual machines, as well as indices and completion time

**Table 2 Summary of notations and their descriptions.**

| Notation | Description |
|---|---|
| $n$ | Number of programs. |
| $J$ | Set of $n$ distinct programs. |
| $m$ | Number of virtual machines. |
| $j$ | Index of the program. |
| $i$ | Index of virtual machine. |
| $VM$ | Set of virtual machines. |
| $p_j$ | Processing duration of the program $j$. |
| $t_j$ | Total time expended on the processing of program $j$ once it has been allocated |
| $C_i$ | Overall execution time of virtual machine $i$. |
| $C_{max}$ | Uppermost completion times upon the successful finishing of all scheduling programs. |
| $C_{min}$ | Lowest completion times upon the successful finishing of all scheduling programs. |

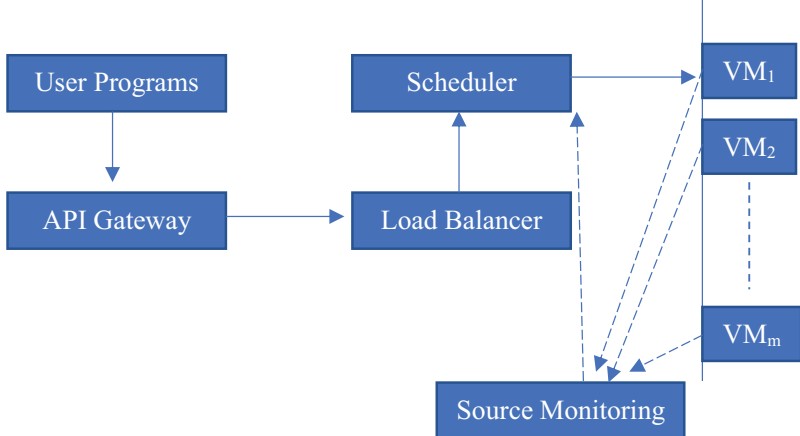

**Figure 1 Cloud-native workflow architecture.** This structure demonstrates the relationships among system elements in a cloud virtualized environment. Programs are submitted by the user *via* an API gateway, which then sends the requests to a load balancer. Load balancer: It, in cooperation with the scheduler, balances the tasks on multiple VMs ($VM_1$ to $VM_m$) of the same capacity. A source monitoring module that continuously monitors the VM's performance metrics and reports it back to the load balancer and scheduler module. Solid arrows denote dominant control and task flow, dotted arrows indicate monitoring and feedback communication connections.

metrics. These notations underpin the proposed workload scheduling model and its ensuing analysis.

## Problem presentation

Let the set $J$ a specific of $n$ distinct programs that are to be systematically organized and appropriately scheduled across $m$ parallel virtual machines that operate simultaneously. Each program, designated as $j$, possesses and is characterized by a unique set of attributes that dictate its operational requirements and constraints within the scheduling framework.

**Table 3 Six programs and two virtual machine instances.**

| $j$ | 1 | 2 | 3 | 4 | 5 | 6 |
|---|---|---|---|---|---|---|
| $p_j$ | 9 | 12 | 8 | 7 | 11 | 5 |

The diagram depicted in Fig. 1 meticulously delineates five fundamental components that are crucial to the overall architecture, which can be outlined as follows:

- API gateway: This component serves as the primary interface through which all incoming program submissions are received and processed, effectively acting as the first line of interaction between external requests and the internal architecture.
- Load balancer: This fundamental system is important for spreading out incoming requests across various schedulers, guaranteeing that these requests are managed in a proficient and coordinated fashion without triggering any delays or holdups.
- Scheduler: Operating as a crucial manager, the scheduler is in charge of the thoughtful task assignment to diverse virtual machines, or $VM$s, and it skillfully directs the execution of these tasks to ensure superior performance.
- $VM$ instances ($VM_1, VM_2, VM_3, \ldots, VM_m$): These virtual machine instances represent the actual computational environments where programs are executed, providing the necessary infrastructure for running applications in a cloud-native ecosystem.
- Resource monitor: This sophisticated component is tasked with continuously collecting real-time metrics, including, but not limited to, CPU usage, memory consumption, and overall load from each virtual machine. Subsequently, it relays this information back to the scheduler through a dashed feedback loop, which facilitates dynamic adjustments to resource allocation.

This architectural framework epitomizes a quintessential cloud-native workflow and vividly illustrates the cooperative dynamics of monitoring and orchestration that work in tandem to uphold both performance standards and fairness across the various components involved. Should you seek a more comprehensive investigation into specific elements, such as a container orchestrator like Kubernetes, a dedicated datastore, or the subtleties of network layers, please inform me for more clarity.

**Example 1** *The number of available programs for utilization is denoted as six, while the number of virtual machines at disposal is established as two. This particular example is solely focused on the scheduling of programs that are intended to be executed on the initial day of operations. Moving forward to the subsequent working day, a fresh assortment of programs is to be allocated for execution across the two virtual machines that have been previously identified. In this context, Table 3 provides a comprehensive enumeration of the various $p_j$ values associated with the aforementioned programs.*

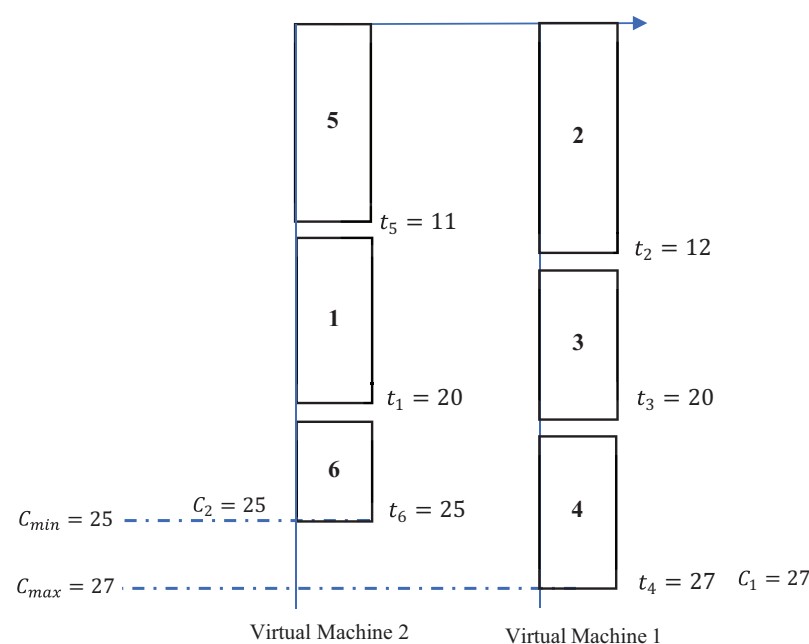

**Figure 2 Illustration of a dispatching rule example.** The distribution of six jobs in two virtual machines by invoking a dispatching rule for every job $i$, there is a rectangle such that $i$ is inside the rectangle, and the length of the rectangle is $t_i$. Jobs are scheduled to VMs according to the schedule rule that aims to equilibrium the makespan. VM1 applies the tasks 2, 3, and 4, having completion time $C_1 = 27$ and VM2 does the tasks 5, 1, and 6, having completion time $C_2 = 25$. The values $C_{min} = 25$, $C_{max} = 27$ are considered. This figure illustrates how task sequencing and VM selection influence load balance and fairness.

The primary objective of this research endeavor is to meticulously explore and identify a comprehensive scheduling framework that will effectively allocate and assign all specified programs across the two designated virtual machines that have been provided for analysis. When the heuristic known as the largest processing-time-first (commonly abbreviated as LPT) is implemented as a dispatching rule for optimal scheduling, the resultant schedule is depicted and illustrated in the graphical representation shown in *Fig. 2*.

The LPT dispatching rule, as illustrated in the accompanying *Fig. 2*, delineates a systematic approach whereby, on the initial virtual machine designated as virtual machine 1, there is a scheduled allocation of the programs identified as $\{2, 3, 4\}$, while concurrently, on the subsequent virtual machine referred to as virtual machine 2, there exists an assignment of the programs categorized as $\{5, 1, 6\}$. Upon thorough examination of *Fig. 1*, it can be ascertained that the aggregate processing time for the first virtual machine totals an impressive 27 units, in contrast to the second virtual machine, which exhibits a total completion time that amounts to 25 units. Consequently, the disparity that exists between the completion times associated with virtual machine 1 and its counterpart, virtual machine 2, can be mathematically represented as $C_{max} - C_{min} = 2$, indicating a clearly defined gap of 2 units. The primary objective underpinning the research and findings articulated within this work is fundamentally centered on the aspiration to minimize this identified gap. Thus,

*it becomes imperative to explore and establish a more effective assignment strategy that achieves a gap measurement that is quantitatively less than the current threshold of 2 units.*

## System assumptions and applicability

In this research, the proposed framework operates based on a defined set of explicit assumptions to ensure both clarity and applicability. The context examined is a virtualized cloud computing infrastructure, wherein a finite collection of independent workloads (or applications) must be allocated to a selection of VMs. It is argued that the workloads are not subject to interruption, implying that once an application is put on a VM, it will run to its end without any breaks. Each workload possesses a known and deterministic processing duration ($p_j$), and all VMs are homogeneous in their execution capabilities unless stated otherwise. Furthermore, there are no interdependencies among workloads, and communication or migration overheads are regarded as negligible. These assumptions accurately represent numerous practical scenarios within cloud and data center environments where task scheduling and resource allocation are of paramount importance. Even though the model is built for consistent scenarios, it can be adjusted to meet fluctuating or irregular workloads *via* additional tools established to confront uncertainties.

This research presumes that all VMs within the system exhibit homogeneity, signifying that they possess identical computational power, memory, and storage capabilities. This assumption facilitates a more precise examination of the fairness and workload balancing mechanisms proposed in the proposed heuristics. Nonetheless, it is recognized that in practical cloud environments, VMs frequently display heterogeneity, characterized by differing central processing unit (CPU) speeds, graphics processing unit (GPU) capabilities, and memory limitations. The existing heuristics can be modified for heterogeneous environments by integrating weights or normalization factors that reflect VM capabilities during the workload allocation process. For example, processing times $p_j$ could be calibrated in relation to VM performance, and the assignment probabilities in the proposed probabilistic methodologies could be adjusted to prioritize VMs with superior capacities for resource-intensive tasks. Expanding the heuristics to accommodate such heterogeneity presents a promising avenue for future work.

In this research, it is assumed that all tasks possess fixed and deterministic processing durations $p_j$ and that the complete workload is predetermined. This assumption of a static workload enables the assessment of the efficacy of the proposed heuristics in reducing completion time disparities under controlled conditions. Nevertheless, it is recognized that actual cloud environments frequently display dynamic characteristics, encompassing varying task priorities, real-time influx of new workloads, and possible task cancellations. The proposed heuristics can be tailored to accommodate such dynamic settings by incorporating online scheduling mechanisms that progressively adjust allocations in response to variations in workload. For instance, the probabilistic assignment phase could be refined to include newly arriving tasks without disrupting existing assignments, while ensuring fairness and load balancing among virtual machines. Evaluating and validating

the heuristics within dynamic workload contexts represents a promising avenue for future work.

## Mathematical model

To thoroughly assess and analyze the disparities that exist among the various virtual machines, a comprehensive selection of pertinent indicators has been meticulously identified and chosen for this evaluative process. Within the boundaries of this research document, the distinct parameter that is presented for review and detailed investigation is conveyed mathematically as the distinction between the utmost completion timeframe, signified as $C_{max}$, and the least completion timeframe, indicated as $C_{min}$. When the entire set of virtual machines under investigation, the overall gap in total completion times is quantitatively represented in the mathematical formulation provided in Eq. (1).

$$\min \sum_{k=1}^{m} [C_k - C_{min}]. \tag{1}$$

In Eq. (1), the variation for a designated value of $k$ is labeled $dif_k$, which is mathematically delineated as the difference between $C_k$ and $C_{min}$. Consequently, it follows that Eq. (1) can be reformulated in the following manner: $\min \sum_{k=1}^{m} dif_k$. This aggregate is referred to as $gap$, which is mathematically represented by the summation $gap = \sum_{k=1}^{m} dif_k$. The problem under investigation is categorized as NP-hard, thereby indicating that the objective is to devise a scheduling strategy that effectively minimizes the cumulative discrepancies, or gaps, that exist between the virtual machine exhibiting the least total completion time and all other virtual machines within the system.

**Proposition 1** *The summation of all running time for all programs can be written as given in Eq. (2).*

$$\sum_{j=1}^{n} p_j = \sum_{k=1}^{m} C_k. \tag{2}$$

**Proof 1** *Denote, for the purpose of clarity and specificity, by the notation $J_k$ the comprehensive set of all computing programs that are executed within the operational confines of a designated virtual machine denoted by $k$. Therefore, the aggregate summation of the processing time associated with each individual program contained within the set $J_k$ can be articulated mathematically as $\sum_{j \in J_k} p_j$, which is equivalently expressed as $C_k$, representing the total computational time consumed by the programs executed by this particular virtual machine. Conversely, it is imperative to consider that all programs that are executed by the virtual machine denoted by $k$, given the condition that*

$$\sum_{j=1}^{n} p_j = \sum_{k=1}^{m} \sum_{j \in J_k} p_j. \tag{3}$$

*Finally, Eq. (2) is proven.*

**Remark 1** *Equation (2) serves as a pivotal mechanism for rigorously assessing the precision and reliability of the algorithms that have been proposed for implementation. Indeed, each individual outcome that is derived from any given algorithm undergoes a comprehensive evaluation through the meticulous calculation of the summation denoted by $\sum_{k=1}^{m} C_k$, which is subsequently returned by the algorithm in question. Should it be determined that this specific summation aligns with the summation denoted by $\sum_{j=1}^{n} p_j$, one may deduce that the reliability of the suggested algorithm complies with the recognized benchmarks set forth in the field. Conversely, if the aforementioned condition is not satisfied and discrepancies are identified, it signifies the presence of an error, thereby necessitating the rejection of the results produced by the algorithm.*

**Example 2** *Suppose that a corporation engages in the operation of four identical virtual machines concurrently and in unison, thereby maximizing their computational capabilities. The first virtual machine necessitates an immediate intervention to replace its worn components on the date of April 17th, 2025, at precisely 17:50 h, and subsequently, the second virtual machine similarly requires an intervention for replacement of parts on April 25th, 2025, at 19:21 h. Therefore, a specific duration of exactly 8 days, 1 h, and 31 min divides the moment of the first intervention from that of the following intervention. The primary objective of this research endeavor is to significantly minimize the duration of such temporal gaps that occur between the necessary interventions. In this article, a variety of heuristic approaches are put forth to provide viable solutions to the complex problem under investigation, fundamentally grounded in the principles of the probabilistic method.*

## SCHEDULING ALGORITHMS

This section delineates and articulates two primary heuristics that are intricately associated with the problem that is presently under investigation and analysis. The overarching goal of this endeavor is to elucidate and present approximate solutions that pertain to the NP-hard problem, while simultaneously suggesting avenues for future research that could leverage these heuristics to formulate an exact solution through the establishment of certain upper bounds that govern the problem at hand. The heuristics that have been proposed are fundamentally grounded in the probabilistic method, characterized by various distinct variants that significantly influence their application. These variants encompass the selection of the specific manner in which the iterative approach is systematically employed throughout the analytical process. The extant literature extensively documents the formulation and development of a multitude of algorithms aimed at addressing real-world applications, all of which are fundamentally rooted in the principles of the randomization approach, as evidenced by the works cited (*Ghaderi, 2016*; *Liu & Cheng, 2017*; *Zheng, Wang & Zhang, 2016*; *Agustín et al., 2016*). To effectively tackle the real-world problem that is the focal point of this article, a probabilistic method is put forth, meticulously derived from the foundational principles of the randomization approach.

---

**Algorithm 1** Probabilistic heuristic $(PH_\beta)$.

**Require:** Set of programs $J$, number of programs $n$, processing times $p_j$ for each $j \in J$, set of virtual machines $VM$
**Ensure:** Minimum gap value $gap$
1: Sort all programs in $J$ by non-increasing order of $p_j$
2: **for** $\beta \leftarrow 0.1$ to $0.9$ step $0.1$ **do**
3:     Set $l \leftarrow 1$, $\tilde{J} \leftarrow J$
4:     **while** $l \leq n$ **do**
5:         Generate $\alpha$ randomly in $[1,100]$
6:         **if** $\alpha \leq \beta \times 100$ **then**
7:           Select the first program $J01$ from $\tilde{J}$
8:         **else**
9:           Select the second program $J02$ from $\tilde{J}$
10:         **end if**
11:         Call ASSIGN $(J0x)$ to allocate selected program to a VM
12:         Update $\tilde{J} \leftarrow \tilde{J} \setminus J0x$
13:         Increment $l \leftarrow l + 1$
14:     **end while**
15:     Compute $gap_\beta$ for the current $\beta$
16: **end for**
17: Set $gap \leftarrow \min_{0.1 \leq \beta \leq 0.9} gap_\beta$
18: **return** $gap$

---

## Probabilistic heuristic

The proposed probabilistic heuristic $(PH_\beta)$ is designed to allocate workloads across VMs in a manner that minimizes disparities in their completion times. The algorithm systematically investigates various probabilistic assignment strategies by adjusting a parameter $\beta$, which governs the level of randomness in task selection.

Initially, all programs are ordered in non-increasing sequence based on their processing times $p_j$, thereby prioritizing larger workloads that exert a significant influence on the overall system equilibrium. Within the boundaries of $[0.1, 0.9]$, for each $\beta$, the algorithm executes a loop internally where it probabilistically assigns programs. With each round, a random digit $au$ is drawn evenly from the boundaries $[1,100]$. In the event that $\beta imes 100$ encompasses $\beta$ values from 1 to $\beta imes 100$, the program that exhibits the longest processing duration from the leftover group $ildeJ$ is selected; if not, the program with the next longest processing duration is chosen. This probabilistic decision-making introduces a controlled element of randomness, enabling the algorithm to circumvent suboptimal deterministic assignment patterns.

Once a program is selected, the ASSIGN procedure allocates it to an appropriate VM, updates the VM's load, and removes the program from the set $\tilde{J}$. After the allocation of all programs, the algorithm calculates the gap $gap_\beta$, which quantifies the disparity between the maximum and minimum completion times across all VMs for the current value of $\beta$. This procedure is reiterated for each $\beta$, with the minimum observed gap being returned as the final outcome.

The parameter $\beta$ is instrumental in balancing exploitation and exploration. Lower values of $\beta$ favor deterministic selections of the largest workloads, whereas higher values of $\beta$ introduce greater randomness, which may reveal more advantageous workload

---

**Algorithm 2** Repetitive probabilistic heuristic ($PH_{\beta}^{lm}$).

**Require:** Number of iterations $lm$, set of programs $J$, processing times $p_j$ for each $j \in J$, set of virtual machines $VM$
**Ensure:** Minimum gap value $g$
1: **for** $k \leftarrow 1$ to $lm$ **do**
2:     Execute $PH_{\beta}$ to obtain $gap_k$
3: **end for**
4: Compute $g \leftarrow \min_{1 \leq k \leq lm} gap_k$
5: **return** $g$

---

distributions. By examining multiple values of $\beta$, $PH_{\beta}$ determines the most efficient probabilistic strategy for achieving an optimal balance in system load.

The probabilistic heuristic ($PH_{\beta}$) is illustrated in Algorithm 1.

**Proposition 2** *The probabilistic algorithm PH running with a complexity of $O(n\log n)$.*

**Proof 2** *The sorting mechanism that is employed within the confines of the algorithmic framework in question is characterized as heapsort, which is a well-known and widely studied sorting technique in computer science. In computational terms, heapsort is defined as an algorithm that has a time complexity of $O(n \log n)$, pointing out that its effectiveness is logarithmically linked to the input data size, marked by n. Also, it is imperative to indicate that an aggregate of $n - 1$ random numerical outputs are created, and based on the specific application that is preferred, the related computational assignments are distributed as necessary; this whole procedure is executed within a time complexity of $O(n)$. The various programs that are contained within each specific set are systematically assigned to a virtual machine, which inherently requires a time complexity of $O(n)$ for their execution. The process involves the repetition of the execution for each distinct value of $\beta$, leading to a comprehensive analysis of performance across these parameters. In this context, a total of 9 discrete values is considered. It is also significant to assert that n is notably much greater than 9, represented mathematically as $n \gg 9$. Consequently, the overall complexity associated with the PH heuristic is determined to be $O(n \log n)$, aligning with the established theoretical understanding of its performance characteristics.*

## Repetitive probabilistic heuristic

The iterative probabilistic heuristic ($PH_{\beta}^{lm}$) enhances the foundational $PH_{\beta}$ algorithm by performing it multiple times to improve both the quality and dependability of the solution. This methodology seeks to mitigate the variability introduced by randomization and to ascertain a more resilient allocation strategy.

The algorithm accepts the number of iterations $lm$ as input. For each iteration $k \in \{1, 2, \ldots, lm\}$, the $PH_{\beta}$ algorithm is executed a single time, yielding a gap value $gap_k$. Each $gap_k$ quantifies the difference between the maximum and minimum completion times across all virtual machines for that specific iteration. Upon the conclusion of all $lm$ iterations, the algorithm identifies the smallest gap $g = \min_{1 \leq k \leq lm} gap_k$ as the conclusive result and returns it.

Through the repeated execution of the probabilistic heuristic, $PH_{\beta}^{lm}$ amplifies the probability of discerning a superior workload distribution that minimizes disparities in completion times. This iterative process is particularly efficacious in counterbalancing the stochastic nature of $PH_{\beta}$, thereby ensuring that the final solution exhibits diminished sensitivity to the random decisions made during task allocation.

In practical applications, it is feasible to configure the value of $lm$ to be equal to 500, thereby establishing a robust framework for the iterative process and ensuring the heuristic explores a wide solution space to minimize completion time disparities effectively.

The Repetitive probabilistic heuristic ($PH_{\beta}^{lm}$) is illustrated in Algorithm 2.

**Proposition 3** *The repetitive probabilistic algorithm $PH_{\beta}^{lm}$ running with a complexity of $O(n^2)$.*

**Proof 3** *The sorting procedure that is implemented within the confines of this sophisticated algorithm is none other than the heapsort, which is renowned for its efficiency and effectiveness in systematically arranging data. The heapsort procedure, which is categorized as an $O(n \log n)$ algorithm, signifies that the time complexity grows logarithmically with the number of elements, thereby ensuring optimal performance even as the dataset expands in size. Furthermore, it is crucial to note that a total of $n - 1$ random numbers are provided as input, and the assignment of jobs is determined based on the specific job that has been selected; under these circumstances, this particular procedure operates within a time complexity of $O(n)$. The various jobs that are grouped together in each set are subsequently scheduled to be executed on a virtual machine, which itself necessitates a time complexity of $O(n)$ for proper execution and management of the tasks at hand. The execution of this process is repeated a total of 500 times, and this repetition is recorded as maintaining a consistent running time order proportional to $n$. Considering these points, one can definitely conclude that the complexity linked to the PH heuristic is defined by an $O(n \log n)$ computational complexity.*

### Mixed-probabilistic heuristic

The mixed-probabilistic heuristic (MPH) presents an advanced task ordering methodology designed to optimize workload distribution among virtual machines. This arrangement results in the programs $J$ being partitioned into two equivalent sections. The initial subset, $J_1$, which consists of 50% of the programs, is organized in a non-increasing sequence based on their processing times $p_j$. This arrangement prioritizes larger workloads at the outset of the assignment procedure. In contrast, the second subset, $J_2$, which encompasses the remaining 50% of programs, is arranged in a non-decreasing order of $p_j$, thereby postponing the assignment of smaller workloads. Upon joining $J_1$ and $J_2$ to produce the restructured group $J$, a chance-based approach is utilized to support task distribution. Should the original probabilistic heuristic $PH_{\beta}$ be utilized, the resultant algorithm is designated as $MPH_{\beta}$. In another approach, employing the repetitive probabilistic heuristic $PH_{\beta}^{lm}$ leads to designate the resultant algorithm as $MPH_{\beta}^{lm}$. This dual-phase sorting strategy introduces variation in task ordering, enabling the heuristic to effectively balance the prompt allocation of substantial workloads with the delayed

---

**Algorithm 3  Mixed-probabilistic heuristic (MPH).**

**Require:** Set of programs $J$, number of programs $n$, processing times $p_j$ for each $j \in J$, set of virtual machines $VM$, heuristic type ($PH_\beta$ or $RPH(\beta, lm)$)
**Ensure:** Workload assignment minimizing completion time disparities
1: Divide $J$ into two subsets: $J_1$ and $J_2$ such that $|J_1| = |J_2| = n/2$
2: Sort $J_1$ in **non-increasing** order of $p_j$
3: Sort $J_2$ in **non-decreasing** order of $p_j$
4: Concatenate $J = J_1 \cup J_2$
5: **if** using $PH_\beta$ **then**
6:     Execute $PH_\beta$ on reordered set $J$
7: **else if** using $PH_\beta^{lm}$ **then**
8:     Execute $PH_\beta^{lm}$ on reordered set $J$
9: **end if**
10: **return** workload assignment and associated gap

---

**Algorithm 4  Reversed mixed-probabilistic heuristic (RMPH).**

**Require:** Set of programs $J$, number of programs $n$, processing times $p_j$ for each $j \in J$, set of virtual machines $VM$, heuristic type ($PH_\beta$ or $PH_\beta^{lm}$)
**Ensure:** Workload assignment minimizing completion time disparities
1: Divide $J$ into two subsets: $J_1$ and $J_2$ such that $|J_1| = |J_2| = n/2$
2: Sort $J_1$ in **non-decreasing** order of $p_j$
3: Sort $J_2$ in **non-increasing** order of $p_j$
4: Concatenate $J = J_1 \cup J_2$
5: **if** using $PH_\beta$ **then**
6:     Execute $PH_\beta$ on reordered set $J$
7: **else if** using $PH_\beta^{lm}$ **then**
8:     Execute $PH_\beta^{lm}$ on reordered set $J$
9: **end if**
10: **return** workload assignment and associated gap

---

assignment of lighter ones, thereby enhancing the equity of the resultant workload distribution.

The mixed-probabilistic heuristic is illustrated in Algorithm 3.

## Reverse-mixed probabilistic heuristic

The reversed mixed-probabilistic heuristic (RMPH) represents a sophisticated adaptation of the mixed-probabilistic strategy, specifically crafted to implement an alternative mechanism for task ordering in workload distribution. This system necessitates a first division of the array of programs $J$ into two uniform portions, $J_1$ and $J_2$. The first subset, $J_1$, is organized in a non-decreasing sequence based on the processing times $p_j$, thereby facilitating the precedence of allocating lighter workloads at the outset of the process. In contrast, the second subset, $J_2$, is arranged in a non-increasing order of $p_j$, thus ensuring that heavier workloads are postponed to the subsequent phases of allocation. As $J_1$ and $J_2$ are integrated to develop the reordered collection $J$, a probabilistic strategy is implemented to allocate the workloads to the virtual machines.

Should the original probabilistic heuristic $PH_\beta$ be utilized, the resultant algorithm is designated as $RMPH_\beta$. Conversely, if the repetitive probabilistic heuristic $PH_\beta^{lm}$ is applied, the resulting algorithm is termed $RMPH_\beta^{lm}$. This reversed ordering methodology

engenders a distinct allocation dynamic in comparison to *MPH*, thereby facilitating a more extensive exploration of potential workload distributions and offering an additional strategy to mitigate completion time disparities among virtual machines.

The reversed mixed-probabilistic heuristic is illustrated in Algorithm 4.

## TEST AND EXPERIMENTS

In this section, a detailed discourse concerning the proposed heuristics is thoroughly articulated, utilizing a diverse array of metrics and instances to enhance a comprehensive understanding of their efficacy and relevance. A systematic and detailed comparison will be conducted among all the proposed heuristics to elucidate their relative performance and characteristics in a clear and precise manner. In order to implement all proposed heuristics outlined in this study, Microsoft Visual C++ was used, which serves as the foundational coding platform for the computational experiments. The computational system utilized for this rigorous research endeavor is an Intel(R) Core (TM) i5-3337U CPU, providing the necessary processing power to execute the algorithms efficiently and effectively.

### Instances

In the researches referenced in *Eljack et al. (2024, 2023)*, a variety of distinct instances are meticulously presented, which are derived from several categorized classes that have been rigorously analyzed. The study specifically focuses on two primary types of statistical distributions that are under consideration, namely the uniform distribution, which is represented symbolically as *UD*[], and the normal distribution, which is denoted by the notation *ND*[]. Through this comprehensive examination, the research endeavors to elucidate the characteristics and implications of these two fundamental classes within the broader context of statistical analysis and its applications.

The specified duration that is necessary for the execution of a particular computational program, denoted as $j$, which is represented mathematically as $p_j$, has been articulated in the following manner:

- Class $C_1$: $p_j \in UD[1, 100]$.
- Class $C_2$: $p_j \in UD[10, 150]$.
- Class $C_3$: $p_j \in UD[100, 500]$.
- Class $C_4$: $p_j \in ND[50, 100]$.
- Class $C_5$: $p_j \in ND[25, 100]$.

Two categories of instances were generated: Small Instances and Big Instances. The $(n, m)$ configurations for the small instances are summarized in Table 4. Moreover, the configurations for the big instances are summarized in Table 5.

In relation to each individual value represented by the triplet $(n, m, class)$, a total of ten distinct instances were systematically generated to ensure a comprehensive exploration of the parameter space. Upon analyzing the information presented in Table 4, it can be deduced with a high degree of certainty that the aggregate total of all instances produced amounts to a substantial figure of 1,250 for the small instances.

**Table 4** $(n, m)$ **choices for small instances.**

| $n$ | $m$ |
| --- | --- |
| 10, 15, 20 | 2, 4, 5 |
| 60, 100, 120, 200 | 2, 4, 10, 15 |

**Table 5** $(n, m)$ **choices for big instances.**

| $n$ | $m$ |
| --- | --- |
| 500, 1,000, 1,500, 2,000, 2,500 | 50, 100, 150, 200 |

Based on Table 5, a total of 1,000 big instances was generated. In addition, 1,250 small instances were produced (see Table 4), bringing the overall total to 2,250 instances.

## Metrics

In this article, the following metrics were used:

- $A_b$: The best heuristic value is found after the execution of all heuristics.
- $A$: the value of the heuristic suggested
- $G_b = \frac{A - A_b}{A}$: the distance between the minimum heuristic value and the given one.
- $Time$: average running time in seconds. Running times below 0.001 s are indicated "-".
- $Per$: fraction of files where $A_b = A$ among all the 1,250 cases.

## Performance analysis on small instances

This subsection presents a detailed analysis of the results obtained from the 1,250 small instances. All statistical evaluations discussed herein are based solely on this subset of instances.

Table 6 provides a comprehensive illustration demonstrating that the heuristic denoted as $PH_\beta^{lm}$ not only stands out as the most effective option but also yields the most favorable gap, achieving a performance percentage of $Per = 82.2\%$, while concurrently maintaining an average gap value of 0.11, all accomplished within a remarkably brief average processing time of 0.036 s. In stark contrast, the heuristic identified as $MPH_\beta$ emerges as the least effective choice, as it produces the most significant gap, registering a performance percentage of merely $Per = 4.4\%$, and accomplishing this in a time frame that is less than 0.001 s, alongside a gap value represented as $G_b$ equal to 0.83.

Table 7 provides a comprehensive presentation of the variations in both the gap denoted as $G_b$ and the corresponding execution time referred to as $Time$, all in relation to the varying parameter $n$. This detailed analysis clearly demonstrates that the algorithm designated as $MPH_\beta$ achieves an impressive maximum value of the gap, which is recorded at an exceptional level of 0.94, specifically when the parameter $n$ is set to 120. Conversely, it is noteworthy to mention that the most favorable gap, characterized by a remarkably low value of less than 0.01, is realized through the implementation of the algorithm $PH_\beta^{lm}$.

**Table 6 Overall performance of all algorithms for small instances.**

| | $PH_\beta$ | $PH_\beta^{lm}$ | $MPH_\beta$ | $MPH_\beta^{lm}$ | $RMPH_\beta$ | $RMPH_\beta^{lm}$ |
|---|---|---|---|---|---|---|
| *Perc* | 29.5% | 82.2% | 4.4% | 28.6% | 20.5% | 59.0% |
| $G_b$ | 0.48 | 0.11 | 0.83 | 0.59 | 0.61 | 0.28 |
| *Time* | – | 0.036 | – | 0.039 | – | 0.034 |

**Table 7 Variation of $G_b$ and execution time as a function of $n$ for small instances.**

| $n$ | $PH_\beta$ | | $PH_\beta^{lm}$ | | $MPH_\beta$ | | $MPH_\beta^{lm}$ | | $RMPH_\beta$ | | $RMPH_\beta^{lm}$ | |
|---|---|---|---|---|---|---|---|---|---|---|---|---|
| | $G_b$ | *Time* | $G_b$ | *Time* | $G_b$ | *Time* | $G_b$ | *Time* | $G_b$ | *Time* | $G_b$ | *Time* |
| 10 | 0.29 | – | 0.00 | 0.003 | 0.60 | – | 0.31 | 0.004 | 0.52 | – | 0.26 | 0.003 |
| 15 | 0.56 | – | 0.03 | 0.004 | 0.75 | – | 0.48 | 0.004 | 0.73 | – | 0.39 | 0.004 |
| 20 | 0.69 | – | 0.04 | 0.006 | 0.87 | – | 0.57 | 0.005 | 0.78 | – | 0.37 | 0.006 |
| 60 | 0.58 | – | 0.18 | 0.021 | 0.90 | – | 0.68 | 0.021 | 0.63 | – | 0.30 | 0.019 |
| 100 | 0.43 | – | 0.11 | 0.040 | 0.82 | – | 0.61 | 0.042 | 0.65 | – | 0.41 | 0.037 |
| 120 | 0.52 | – | 0.24 | 0.051 | 0.94 | – | 0.71 | 0.060 | 0.48 | – | 0.05 | 0.049 |
| 200 | 0.32 | – | 0.12 | 0.105 | 0.87 | – | 0.66 | 0.109 | 0.51 | – | 0.24 | 0.101 |

Furthermore, it is significant to highlight that the algorithm $PH_\beta^{lm}$ reaches its peak execution time of 0.105 s precisely when the value of $n$ is increased to 200.

Table 8 presents an extensive analysis of the various values associated with $G_b$ and *Time* that emerge as a consequence of alterations in the quantity of support storage, thereby illustrating the intricate relationships between these parameters. Upon reviewing the data illustrated in the table, it is evident that the most beneficial gap is achieved through the algorithm identified as $PH_\beta^{lm}$ when the variable $m$ equals 2, while the algorithm $MPH_\beta^{lm}$ results in the least favorable gap, noted at 0.95, when the variable $m$ is heightened to 10. In the context of the heuristics denoted as $PH_\beta$, $MPH_\beta$, and $RMPH_\beta$, it is noteworthy that the mean duration of execution remains consistently below 0.001 s, indicating a remarkable efficiency in their operational performance. In stark contrast, the algorithm identified as $MPH_\beta^{lm}$ demonstrates a significantly longer average execution time that can escalate to 0.072 s when the parameter $m$ is elevated to the value of 15, thereby highlighting a marked discrepancy in computational efficiency between the various algorithms under consideration.

Table 9 meticulously delineates the comprehensive results that elucidate the fluctuations observed in both $G_b$ and *Time* as a function of the alterations in class categories. This table explicitly indicates that classes 4 and 5 present a significantly greater level of difficulty in comparison to the other classes for the algorithms $PH_\beta$, $PH_\beta^{lm}$, and $MPH_\beta$, as evidenced by the fact that the average gap yields notably higher numerical values for these specific class categories. Conversely, with regard to the algorithms $MPH_\beta^{lm}$, $RMPH_\beta$, and $RMPH_\beta^{lm}$, there is a discernible reduction in the average gap for classes 4 and 5 when juxtaposed with the other class categories, suggesting a contrasting performance outcome.

**Table 8 Variation of $G_b$ and execution time as a function of $m$ for small instances.**

| $m$ | $PH_\beta$ | | $PH_\beta$ | | $MPH_\beta$ | | $MPH_\beta^{lm}$ | | $RMPH_\beta$ | | $RMPH_\beta^{lm}$ | |
|---|---|---|---|---|---|---|---|---|---|---|---|---|
| | $G_b$ | Time | $G_b$ | Time | $G_b$ | Time | $G_b$ | Time | $G_b$ | Time | $G_b$ | Time |
| 2 | 0.43 | – | 0.00 | 0.027 | 0.79 | – | 0.02 | 0.030 | 0.50 | – | 0.06 | 0.028 |
| 4 | 0.54 | – | 0.05 | 0.028 | 0.87 | – | 0.81 | 0.031 | 0.74 | – | 0.50 | 0.028 |
| 5 | 0.38 | – | 0.04 | 0.004 | 0.74 | – | 0.68 | 0.004 | 0.56 | – | 0.36 | 0.004 |
| 10 | 0.65 | – | 0.40 | 0.059 | 0.96 | – | 0.95 | 0.060 | 0.61 | – | 0.12 | 0.054 |
| 15 | 0.36 | – | 0.18 | 0.067 | 0.78 | – | 0.76 | 0.072 | 0.59 | – | 0.40 | 0.061 |

**Table 9 $G_b$ and *Time* variation regarding *Class* for small instances.**

| Class | $PH_\beta$ | | $PH_\beta^{lm}$ | | $MPH_\beta$ | | $MPH_\beta^{lm}$ | | $RMPH_\beta$ | | $RMPH_\beta^{lm}$ | |
|---|---|---|---|---|---|---|---|---|---|---|---|---|
| | $G_b$ | Time | $G_b$ | Time | $G_b$ | Time | $G_b$ | Time | $G_b$ | Time | $G_b$ | Time |
| 1 | 0.30 | – | 0.04 | 0.037 | 0.85 | – | 0.63 | 0.040 | 0.59 | – | 0.32 | 0.035 |
| 2 | 0.44 | – | 0.06 | 0.036 | 0.83 | – | 0.62 | 0.039 | 0.63 | – | 0.32 | 0.034 |
| 3 | 0.53 | – | 0.05 | 0.036 | 0.87 | – | 0.59 | 0.038 | 0.71 | – | 0.33 | 0.034 |
| 4 | 0.54 | – | 0.19 | 0.036 | 0.87 | – | 0.60 | 0.039 | 0.61 | – | 0.26 | 0.035 |
| 5 | 0.59 | – | 0.24 | 0.035 | 0.73 | – | 0.50 | 0.037 | 0.49 | – | 0.18 | 0.034 |

In the comprehensive presentation of the findings, a greater level of detail is meticulously provided in Table 10, which serves to elucidate the performance metrics encompassing both the average gap achieved and the time expended for all heuristics that have been developed throughout this research endeavor.

Figure 3 presents a comprehensive visual representation of the varying values of the average gap in relation to the numerical variable denoted as $nb$, specifically for the heuristic methodologies referred to as $PH_\beta$ and $PH_\beta^{lm}$. This particular figure effectively demonstrates that the graphical representation of the curve associated with the heuristic $PH_\beta$ consistently resides at a higher position on the graph compared to the curve linked to the heuristic $PH_\beta^{lm}$. Therefore, this analysis reveals that the heuristic $PH_\beta^{lm}$ is positioned to offer a more advantageous solution than the heuristic $PH_\beta$ when the assorted values of the parameter $nb$ are considered.

Figure 4 presents a comprehensive depiction of the varying numerical values associated with the average gap in relation to the parameter denoted as $nb$, which is being analyzed for the two distinct heuristics, namely $MPH_\beta$ and $MPH_\beta^{lm}$. This particular figure serves to elucidate the observation that the graphical representation of the heuristic $MPH_\beta$ consistently resides at a higher position on the plotted axis when compared to the graphical representation of the heuristic $MPH_\beta^{lm}$. As a result, this deduction leads to the assertion that the heuristic $MPH_\beta^{lm}$ demonstrates a greater efficacy in producing a more optimal solution relative to the heuristic $MPH_\beta$ when assessed against a range of specified values for the parameter $nb$.

**Table 10** $G_b$ and *Time* variation regarding *nb* for small instances.

| nb | n | m | $PH_\beta$ | | $PH_\beta^{lm}$ | | $MPH_\beta$ | | $MPH_\beta^{lm}$ | | $RMPH_\beta$ | | $PH_\beta^{RMlm}$ | |
|---|---|---|---|---|---|---|---|---|---|---|---|---|---|---|
| | | | $G_b$ | Time | $G_b$ | Time | $G_b$ | Time | $G_b$ | Time | $G_b$ | Time | $G_b$ | Time |
| 1 | 10 | 2 | 0.57 | – | 0 | 0.003 | 0.77 | – | 0.12 | 0.003 | 0.88 | – | 0.34 | 0.003 |
| 2 | | 4 | 0.28 | – | 0.01 | 0.003 | 0.49 | – | 0.32 | 0.005 | 0.56 | – | 0.33 | 0.003 |
| 3 | | 5 | 0.01 | – | 0 | 0.002 | 0.52 | – | 0.48 | 0.003 | 0.12 | – | 0.1 | 0.003 |
| 4 | 15 | 2 | 0.6 | – | 0.01 | 0.004 | 0.68 | – | 0.02 | 0.004 | 0.66 | – | 0.06 | 0.003 |
| 5 | | 4 | 0.63 | – | 0.08 | 0.004 | 0.77 | – | 0.7 | 0.004 | 0.68 | – | 0.37 | 0.004 |
| 6 | | 5 | 0.46 | – | 0 | 0.004 | 0.79 | – | 0.73 | 0.004 | 0.84 | – | 0.75 | 0.005 |
| 7 | 20 | 2 | 0.67 | – | 0 | 0.007 | 0.77 | – | 0 | 0.005 | 0.68 | – | 0 | 0.007 |
| 8 | | 4 | 0.71 | – | 0 | 0.005 | 0.93 | – | 0.86 | 0.005 | 0.94 | – | 0.88 | 0.005 |
| 9 | | 5 | 0.67 | – | 0.12 | 0.005 | 0.9 | – | 0.84 | 0.005 | 0.73 | – | 0.22 | 0.005 |
| 10 | 60 | 2 | 0.45 | – | 0 | 0.015 | 0.82 | – | 0 | 0.017 | 0.46 | – | 0 | 0.015 |
| 11 | | 4 | 0.61 | – | 0 | 0.017 | 0.95 | – | 0.92 | 0.017 | 0.97 | – | 0.92 | 0.016 |
| 12 | | 10 | 0.72 | – | 0.41 | 0.023 | 0.95 | – | 0.93 | 0.023 | 0.66 | – | 0.13 | 0.021 |
| 13 | | 15 | 0.56 | – | 0.32 | 0.027 | 0.89 | – | 0.88 | 0.028 | 0.44 | – | 0.15 | 0.025 |
| 14 | 100 | 2 | 0.31 | – | 0 | 0.031 | 0.77 | – | 0 | 0.034 | 0.37 | – | 0 | 0.032 |
| 15 | | 4 | 0.57 | – | 0 | 0.033 | 0.97 | – | 0.95 | 0.036 | 0.97 | – | 0.94 | 0.032 |
| 16 | | 10 | 0.67 | – | 0.45 | 0.044 | 0.96 | – | 0.95 | 0.044 | 0.64 | – | 0.13 | 0.04 |
| 17 | | 15 | 0.15 | – | 0 | 0.051 | 0.57 | – | 0.53 | 0.055 | 0.6 | – | 0.57 | 0.045 |
| 18 | 120 | 2 | 0.27 | – | 0 | 0.041 | 0.88 | – | 0 | 0.055 | 0.25 | – | 0 | 0.042 |
| 19 | | 4 | 0.54 | – | 0.17 | 0.043 | 0.98 | – | 0.97 | 0.057 | 0.57 | – | 0.05 | 0.043 |
| 20 | | 10 | 0.63 | – | 0.39 | 0.057 | 0.96 | – | 0.95 | 0.06 | 0.58 | – | 0.07 | 0.051 |
| 21 | | 15 | 0.63 | – | 0.39 | 0.064 | 0.95 | – | 0.94 | 0.066 | 0.53 | – | 0.09 | 0.058 |
| 22 | 200 | 2 | 0.15 | – | 0.01 | 0.088 | 0.82 | – | 0 | 0.093 | 0.2 | – | 0 | 0.092 |
| 23 | | 4 | 0.44 | – | 0.12 | 0.092 | 0.98 | – | 0.97 | 0.095 | 0.47 | – | 0.02 | 0.092 |
| 24 | | 10 | 0.6 | – | 0.36 | 0.113 | 0.98 | – | 0.97 | 0.111 | 0.57 | – | 0.14 | 0.104 |
| 25 | | 15 | 0.1 | – | 0 | 0.126 | 0.71 | – | 0.69 | 0.138 | 0.81 | – | 0.79 | 0.116 |

## Performance analysis on big instances

Table 11 presents a comparative evaluation of six algorithms predicated on three performance indicators: solution percentage (*Perc*), average deviation from the optimal known solution ($G_b$), and computational duration (*Time*). The findings show that local improvement variants, $PH_\beta^{lm}$, $MPH_\beta^{lm}$, and $RMPH_\beta^{lm}$, consistently surpass their base versions in solution quality. Notably, $PH_\beta^{lm}$ achieves the highest success rate of 72.6%, with a minimal average gap of 0.17, significantly outperforming $PH_\beta$, which has a 55.5% success rate and a gap of 0.20, despite its faster runtime (0.010 s *vs*. 3.553 s).

On the other hand, the multi-population variants $MPH_\beta$ and $MPH_\beta^{lm}$ demonstrate suboptimal performance regarding *Perc*, achieving only 1.8% and 1.9% success rates, alongside high average gaps approximately 0.80, which suggests a possible misalignment with the problem configuration. In contrast, the randomized multi-population variant

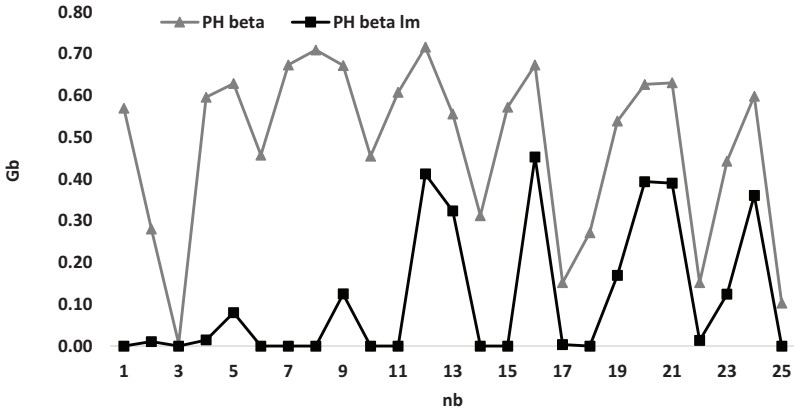

**Figure 3** The average gap with respect to *nb* for $PH_\beta$ and the case $PH_\beta^{lm}$ compared for small instances.

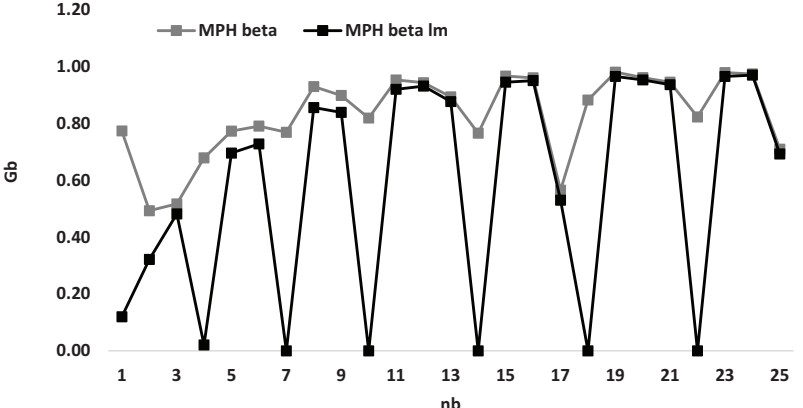

**Figure 4** Average gap according to *nb* for $MPH_\beta$ and $MPH_\beta^{lm}$ for small instances.

**Table 11** Performance evaluation of proposed algorithms in terms of best-value frequency (*Perc*), average gap ($G_b$), and computation time (*Time*).

|  | $PH_\beta$ | $PH_\beta^{lm}$ | $MPH_\beta$ | $MPH_\beta^{lm}$ | $RMPH_\beta$ | $RMPH_\beta^{lm}$ |
|---|---|---|---|---|---|---|
| *Perc* | 55.5% | 72.6% | 1.8% | 1.9% | 17.1% | 36.0% |
| $G_b$ | 0.20 | 0.17 | 0.80 | 0.79 | 0.51 | 0.43 |
| *Time* | 0.010 | 3.553 | 0.009 | 3.292 | 0.010 | 3.504 |

$RMPH_\beta^{lm}$ provides a balanced option with a success rate of 36.0% and a gap of 0.43, while keeping a runtime similar to other local improvement variants (3.504 s).

In summary, $PH_\beta^{lm}$ stands out as the most effective algorithm for solution quality, whereas $PH_\beta$ is the fastest. The data brings to light the gains from incorporating local search frameworks into fundamental algorithms to raise solution quality, even if it does mean more computational time.

**Table 12 Performance comparison of six algorithms with respect to average gap ($G_b$) and running time (*Time*) on different problem sizes $n$.**

| $n$ | $PH_\beta$ | | $PH_\beta^{lm}$ | | $MPH_\beta$ | | $MPH_\beta^{lm}$ | | $RMPH_\beta$ | | $RMPH_\beta^{lm}$ | |
|---|---|---|---|---|---|---|---|---|---|---|---|---|
| | $G_b$ | *Time* | $G_b$ | *Time* | $G_b$ | *Time* | $G_b$ | *Time* | $G_b$ | *Time* | $G_b$ | *Time* |
| 500 | 0.16 | 0.003 | 0.12 | 1.023 | 0.67 | 0.003 | 0.66 | 0.869 | 0.53 | 0.003 | 0.48 | 0.778 |
| 1,000 | 0.26 | 0.005 | 0.22 | 2.263 | 0.78 | 0.005 | 0.77 | 1.935 | 0.47 | 0.006 | 0.36 | 2.281 |
| 1,500 | 0.21 | 0.010 | 0.18 | 3.133 | 0.85 | 0.009 | 0.85 | 2.930 | 0.52 | 0.008 | 0.44 | 3.351 |
| 2,000 | 0.28 | 0.015 | 0.26 | 5.323 | 0.88 | 0.013 | 0.88 | 4.506 | 0.38 | 0.014 | 0.28 | 5.120 |
| 2,500 | 0.09 | 0.019 | 0.07 | 6.020 | 0.80 | 0.018 | 0.80 | 6.219 | 0.62 | 0.018 | 0.57 | 5.991 |

Table 12 presents an extensive comparison of six algorithmic variants—specifically $PH_\beta$, $PH_\beta^{lm}$, $MPH_\beta$, $MPH_\beta^{lm}$, $RMPH_\beta$, and $RMPH_\beta^{lm}$—assessed across various instance classes ($n = 500$ to $2{,}500$). The assessment of performance utilizes two main metrics: the average optimality gap ($G_b$) and the average execution time (*Time* in seconds). The analysis reveals that including local moves, indicated by the "$^{lm}$" suffix, consistently boosts solution quality for every algorithm variant. In particular, $PH_\beta^{lm}$ achieves the lowest average gaps across all instance sizes, attaining a minimum of 0.07 for $n = 2{,}500$, thereby significantly surpassing its foundational counterpart $PH_\beta$. Likewise, the reordered variants, $RMPH_\beta$ and $RMPH_\beta^{lm}$, exhibit competitive performance, with the latter evidently benefiting from local refinement, achieving $G_b = 0.28$ at $n = 2{,}000$ and subsequently improving to $G_b = 0.57$ at $n = 2{,}500$ in contrast to 0.62 in the absence of local moves. Conversely, this enhancement in validity is offset by an extension in computational time. All "$^{lm}$" variants exhibit a considerable increase in execution time, particularly as the problem size escalates—*e.g.*, $MPH_\beta^{lm}$ escalates from 0.869 s ($n = 500$) to over 6 s ($n = 2{,}500$). Conversely, the basic variants devoid of local moves maintain remarkable speed but exhibit noticeably higher optimality gaps. In summary, the locally improved editions deliver higher solution quality albeit with increased computational demands, whereas the more straightforward options ensure scalability and quick processing, thus fitting well in time-critical situations.

## Discussion

The heuristics that have been put forth in this study are fundamentally grounded in the principles of the probabilistic method, which is further enhanced and diversified through the implementation of various distinct variants. These defined variations connect to the unique choice and implementation of the iterative framework, which can be approached in diverse manners relative to the context of the problem at play. The findings from this research decisively confirm the exceptional efficiency and effectiveness of the probabilistic technique, especially when utilized alongside an iterative framework that supports the refinement and enhancement of results. Indeed, it has been rigorously established through comprehensive analysis that the synergistic combination of the probabilistic method alongside the iterative approach yields significantly superior results when juxtaposed with alternative methodologies that may be employed in similar scenarios.

## Trade-off between solution quality and runtime efficiency

Though the considered heuristic algorithms perform well in fairness and load balance, especially when the smallest completion time is desired, it is worth noting that there are some trade-offs between the quality of solutions and the computation time. This can be clearly seen for the multiple repetitions of the heuristics, for example $PH^{lm}\beta$, $MPH^{lm}\beta$ and $RMPH^{lm}_{\beta}$, which in a recurrent way apply the probabilistic assignment to obtain better scheduling results.

While such algorithms provide better solutions by considering a wider range of possible task assignments, they also have higher computational cost with the number of repetitions ($lm$). For the large-scale scenarios having hundreds or thousands of programs and virtual machines, the repeated implementation of the heuristic may have longer running time, which is infeasible for the real time or latency-sensitive tasks in large cloud systems. Also, the current work takes a sequential approach, which is adequate for a simulation study, but it may not be able to utilize available parallelism in cloud or distributed systems. This even more emphasizes the trade-off between complexity of the algorithms and scalability.

To mitigate these issues, future research could explore adaptive stopping criteria, parallel implementations of the heuristics, or the integration of online learning approaches to adjust algorithm parameters on-the-fly and according to the workload properties. These improvements can ensure both solution quality and less computation cost, and make the proposed scheme more feasible in practical cloud scenarios.

## Limitation on task duration assumptions

The scheduler model considered throughout this article is based on the notion that task durations are given *a priori*, indicating that the scheduler operates in a deterministic setting. This assumption allows the problem to be modeled in a less complex way and facilitates a meaningful comparison of the proposed heuristics on instances with different settings. It is recognized that in realistic cloud settings, such dynamism is inherent since task execution times can be unpredictable owing to resource availability fluctuations, system up and down states, and variability of external dependencies. The exclusion of random or variable processing times is thus a limitation of this study. Dealing with this limitation is a promising path for future research, in which integration of predictive modeling, adaptive scheduling, or stochastic optimization methods may enable the potential application of the proposed approach to more realistic cloud computing contexts.

## CONCLUSION

This comprehensive research endeavor has successfully proposed a total of six distinct heuristics subsequent to the establishment of a robust mathematical model that encapsulates the complexities of the problem under investigation. The primary objective of this scholarly work was to significantly diminish the discrepancies observed in the completion times associated with virtual machines, thereby facilitating the standardization of the intervention period within the operational framework of an enterprise. The inaugural category of the algorithms put forward was predicated upon the random selection of a singular task drawn from a pool of the ten largest tasks identified within the

dataset. The second category was developed through the iterative application of a randomized and probabilistic heuristic, executed multiple times to ascertain a singular minimum value that would be optimal for the given context. The third category was constructed utilizing a methodology grounded in mixed ordering principles to enhance the overall effectiveness of the algorithm. A detailed evaluation of the performance benchmarks for the proposed algorithms, especially in connection with time efficiency and gap closure when analyzed against the best-performing algorithm, provided meaningful revelations. The findings unequivocally demonstrate the remarkable efficiency of the algorithms that have been proposed in this study. Notably, the algorithm identified as $PH_{\beta}^{lm}$ emerged as the most effective solution, achieving optimal results in 82.2% of cases, with a minimal gap of 0.11 and an execution time of merely 0.036 s. The innovative solutions articulated within this research may serve as a foundation for future endeavors aimed at deriving precise solutions to the complex problem delineated in this study. Furthermore, the methodologies employed throughout this article to compute the proposed algorithms possess the potential to be adapted for the determination of other algorithms, particularly in diverse metaheuristic contexts. Conversely, it is noteworthy that alternative constraints could be integrated into the problem at hand, thereby facilitating the development of a new NP-hard problem that could pose additional challenges. The practical implications of the problem under investigation within industrial applications could substantially contribute to a significant reduction in maintenance costs, while simultaneously ensuring that the organization can optimize its utilization of virtual machines effectively. In real-world scenarios, the problem presented in this study holds the potential to be generalized and applied to a variety of other domains, including but not limited to healthcare systems, smart parking solutions, hospital management, and network optimization challenges.

### Funding
The authors received no funding for this work.

### Competing Interests
The authors declare that they have no competing interests.

### Author Contributions
- Abdullah Bajahzar conceived and designed the experiments, performed the experiments, analyzed the data, performed the computation work, prepared figures and/or tables, authored or reviewed drafts of the article, and approved the final draft.

### Data Availability
The raw data is available in the Supplemental Files.

## Supplemental Information

Supplemental information for this article can be found online at http://dx.doi.org/10.7717/peerj-cs.3234#supplemental-information.

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
