# Peer review of "Heuristic-based load balancing for identical virtual machines: a fair scheduling approach using probabilistic methods"

_PeerJ Computer Science, doi:10.7717/peerj-cs.3234_

## Round 0.1 · original submission · Major Revisions

**Language Note:** The review process has identified that the English language must be improved. PeerJ can provide language editing services - please contact us at [email protected] for pricing (be sure to provide your manuscript number and title). Alternatively, you should make your own arrangements to improve the language quality and provide details in your response letter. – PeerJ Staff

Reviewer 1 ·

Basic reporting

The paper presents a well-structured and rigorous approach to workload allocation in virtual machines (VMs) using probabilistic heuristics. However, several limitations should be acknowledged to provide a balanced perspective and guide future research:
1. The author should include the improvements found in the proposed work according to the parameters in the abstract. Compared approaches should also be mentioned in the abstract. Too many unnecessary details can be avoided.
2. All the terminology and technology that is used in the title and research work must be introduced in the introduction along with the utility and proper flow that reach the need of the proposed approach.
3. The paper introduces a large number of notations in the modeling process, which appear inconsistent and are very difficult to follow. The presentation of the model should be cleaned up and simplified to improve readability and accessibility for the audience, by collecting all the notations in a table of symbols, for instance.

Experimental design

1. The algorithms are not adequately explained, which makes them challenging to comprehend or replicate. Key parameters and their roles in the model are introduced without sufficient explanation, leaving readers guessing about their significance and usage. Adding structured pseudo-codes, flow diagrams, and clear descriptions of how each step contributes to solving the problem would greatly improve this section.
2. The paper does not address the assumptions underlying the proposed model. For example, it does not specify the type of environment the model is designed for, nor the nature of tasks it supports. These omissions make it unclear whether the model is applicable to a wide range of scenarios or is limited to specific conditions.
3. The study assumes all VMs are identical in terms of computational power, memory, and storage. In real-world cloud environments, VMs are often heterogeneous (e.g., varying CPU speeds, GPU capabilities, or memory constraints). The proposed heuristics may not perform optimally under such conditions.
4. The experiments use fixed task durations (pjpj) and do not account for dynamic workloads (e.g., fluctuating task priorities, real-time arrivals, or cancellations), while cloud environments often experience unpredictable workloads. The algorithms' performance under dynamic conditions remains untested.

Validity of the findings

1. The largest tested instance involves 200 tasks and 15 VMs. Modern cloud systems may scale to thousands of tasks across hundreds of VMs. Needs more experiments to test the scalability.
2. While the paper compares its heuristics internally, it lacks a thorough comparison with the latest algorithms in the domain.
3. The experiments are simulation-based, with no validation on live cloud platforms.

Additional comments

1. Please define abbreviations before their usage. Furthermore, there are some typos errors in the text which leads to ambiguity while understanding the paper. The authors are advised to read the entire manuscript again to avoid such mistakes.
2. All the figures must be of good quality.

Reviewer 2 ·

Basic reporting

-

Experimental design

-

Validity of the findings

-

Additional comments

1. In the scope of this study, it has been stated that the aim was to schedule the tasks of multi-product manufacturing processes on equally capable virtual machines in a balanced manner and that the effectiveness of six proposed heuristic approaches was evaluated through testing on a large number of instances.

2. In the introduction section, industrial enterprise, resource allocation and scheduling, comprehensive research endeavor, and the importance of the subject are clearly mentioned at a sufficient level regarding the study.

3. In the related works section, literature is mentioned to a certain extent regarding balancing process, load-balancing algorithms, and techniques. However, in this section, it would be very useful to add a literature table consisting of certain important parts, such as originality, pros, and cons for this study to stand out more and to reveal its differences from the literature more clearly.

4. When the programs and virtual machine instances, cloud-native workflow architecture, and dispatching rule sections are analyzed in detail regarding the problem definition, it is observed that the problem of this study and the architecture accordingly are clearly explained.

5. Although both the repetitive probabilistic algorithm and the probabilistic algorithm seem to be sufficient for solving the problem in this study, it should be explained more clearly why they were used as the basis since they are compared with other scheduling algorithms in the literature.

6. The types and choices of the instances used in the tests and experiments, and also the types of metrics used, are sufficient. In addition, the results obtained from the algorithms in the study in relation to these metrics are very sufficient when viewed from the perspective of the literature.

As a result, this study is at a certain level. However, it is recommended that all the above sections be explained completely to clearly reveal their contribution to the literature.

Reviewer 3 ·

Basic reporting

-

Experimental design

-

Validity of the findings

-

Additional comments

The article tackles a highly technical issue in scheduling optimization within a virtual ‎manufacturing environment. However, in its current form, it requires significant revisions ‎and improvements. My comments are as follows:‎
• The title should accurately reflect the specific context and methodology of the research. ‎For instance, if the focus is on load balancing across identical parallel virtual machines ‎using heuristic techniques, this should be clearly indicated in the title.‎
• The abstract is overly detailed and fails to concisely summarize the main contributions ‎of the work. The authors are advised to revise it for greater clarity and brevity.‎
• In the introduction, the authors should clearly outline their contributions in a point-by-point format and highlight how these contributions relate to and differ from each other.‎
• The related work section needs expansion. The authors should explain the limitations of ‎existing methods and clearly articulate the novelty of their approach in comparison to ‎the current literature. Including a comparative table would improve clarity and ‎strengthen their analysis. Additionally, the references are somewhat outdated; it is ‎recommended to include more recent studies from 2024–2025, such as:‎
o Adaptive Workload Management in Cloud Computing for Service Level ‎Agreements Compliance and Resource Optimization
o HEPGA: A New Effective Hybrid Algorithm for Scientific Workflow Scheduling in ‎Cloud Computing Environments
o CHPSO: An Efficient Algorithm for Task Scheduling and Optimizing Resource ‎Utilization in the Cloud Environment
o RLSK_US: An Improved Dynamic Virtual Machine Consolidation Model to ‎Optimize Energy and SLA Violations in Cloud Datacenters
• The authors should elaborate on the relevance of their research in current industrial and ‎computational domains such as digital manufacturing, cloud-based production systems, ‎digital twins, and smart factory scheduling.‎
• A deeper discussion of the heuristic algorithm’s potential limitations, especially in terms ‎of computational overhead and scalability, would help assess its practicality in large-scale cloud environments.‎
• The assumption that all virtual machines have identical capabilities and operate ‎concurrently is crucial. This should be explicitly stated, along with a discussion of ‎whether this assumption restricts the applicability of the approach in more realistic, ‎heterogeneous cloud environments.‎
• The article mentions that task durations are predetermined and known in advance, which ‎does not align with real-world cloud environments where such parameters are often ‎uncertain. If the scheduling model is deterministic, the authors should clarify this and ‎acknowledge the exclusion of variable or stochastic processing times as a limitation.‎
• The probabilistic and iterative nature of the proposed framework should be more clearly ‎explained. What probabilistic mechanisms are involved (e.g., random task assignment, ‎randomized neighborhood selection)? What kind of iterative methods are used (e.g., ‎local search, metaheuristics)?‎
• The use of 1,250 benchmark instances is commendable. However, more details are ‎needed regarding the nature of these instances—such as the number of machines, task ‎counts, and duration ranges—and whether they are synthetically generated or based on real-world scenarios.‎
• The manuscript includes several complex or overly technical expressions that may ‎impede understanding. For example:‎
o ‎“Equitable distribution of operational longevity” could be simplified to ‎‎“balanced workload distribution.”‎
o ‎“Facilitating a unified intervention period within an industrial setting” might be ‎rephrased as “ensuring synchronized maintenance or task handovers.”
The authors should consider breaking long, dense sentences into shorter, clearer ‎ones to improve readability and help readers follow the logic without missing ‎important points.

Reviewer 4 ·

Basic reporting

This paper proposed a set of probabilistic heuristics for the fair allocation of workloads across identical virtual machines to minimize disparities in completion times. A mathematical model is formulated, and six heuristic algorithms are developed, including basic, repetitive, and mixed-probabilistic variants. The primary objective is to reduce the gap between the maximum and minimum completion times of virtual machines. Extensive experiments over 1,250 instances demonstrate that the repetitive probabilistic heuristic consistently outperforms others in both efficiency and gap minimization. The results highlight the practical applicability of these heuristics in optimizing resource utilization in cloud computing environments.

This paper deserves consideration due to its novel formulation of the VM scheduling problem, focusing on minimizing disparities in completion times rather than traditional makespan objectives. It introduces a new class of probabilistic heuristics: basic, repetitive, and mixed, that are both innovative and practical, particularly in equitable workload distribution across identical virtual machines. The work addresses a computationally hard and practically relevant problem, offering scalable solutions with strong real-world applicability in cloud environments, and lays the groundwork for future extensions in fairness-aware and energy-efficient scheduling.

Experimental design

The experimental design of the investigation is robust and aligns proficiently with the objectives and scope of the journal, especially in addressing computational methodologies and scheduling optimization within cloud computing environments. The research delineates original primary contributions that introduce probabilistic heuristics aimed at mitigating disparities in virtual machine completion durations, an objective that directly addresses equity in workload distribution, a pivotal yet inadequately examined concern in cloud scheduling.

The research inquiry is distinctly articulated and pertinent, concentrating on the equitable allocation of workloads among homogeneous virtual machines to minimize discrepancies in operational completion times. The authors substantiate their methodology by pinpointing a significant knowledge deficit in the prevailing load-balancing and scheduling literature, wherein the majority of investigations underscore performance indicators such as makespan or energy consumption without adequately addressing fairness.

A meticulous experimental framework is instituted, encompassing the formulation of six heuristics, a formalized mathematical model, and exhaustive testing across 1,250 diverse instances utilizing both controlled and realistic workload distributions. The study adheres to elevated technical standards, employing well-established algorithms (e.g., heapsort), complexity assessments, and performance indicators (e.g., gap, execution duration, success rate) to corroborate the efficacy of the proposed methodologies.

Significantly, the methods are articulated with ample clarity and detail: comprehensive pseudocode, parameter configurations, and implementation specifics (e.g., utilization of Microsoft Visual C++, specifications of the testing system) to ensure complete replicability by other scholars. This guarantees that the research not only contributes novel insights but also upholds transparency and reproducibility in alignment with ethical standards.

The work addresses a computationally hard and practically relevant problem, offering scalable solutions with strong real-world applicability in cloud environments, and lays the groundwork for future extensions in fairness-aware and energy-efficient scheduling.

Validity of the findings

The results delineated in this manuscript are credible and robustly substantiated by comprehensive experimentation and a rigorous methodological framework. Although the significance and originality of the research are not quantitatively evaluated, the introduced probabilistic heuristics provide a novel insight into equitable workload distribution among homogeneous virtual machines—an area that has not been extensively addressed in the prevailing academic discourse. The justification for replication is explicitly articulated, and the manuscript encompasses all requisite data, comprising 1,250 systematically generated instances with diverse parameters, thereby ensuring statistical integrity and reproducibility. The methodologies are articulated with adequate precision, encompassing algorithmic complexity and procedural steps, to facilitate replication and subsequent investigation. The conclusions are systematically inferred from the findings and remain intimately connected to the initial research aim—minimizing inconsistencies in VM completion times—without transcending the confines of the data presented. In summary, the manuscript provides a significant and well-founded contribution to the field of cloud resource scheduling.

Additional comments

The paper is well-written and structured; however, some clarifications and minor revisions must be taken into consideration.

Here are some clarifications:
1. In Table 3, why can’t you see the 100 % in the table? The maximum percentage is 82%, why not 100%?

2. Why is the studied problem NP-hard? Is it proven?

Here are six minor review observations that advocate for acceptance contingent upon minor revisions:

1. Enhance Clarity in the Abstract and Introduction. Although the abstract provides an effective encapsulation of the paper's content, certain sentences exhibit excessive complexity. Simplifying the linguistic constructs and minimizing redundancy (for instance, the reiteration of the objective to "reduce completion time disparities") would significantly enhance the document's readability.

2. Augment Figure Captions. Figures such as Figure 1 and Figure 2 convey valuable information; however, their accompanying captions lack sufficient detail. Incorporating more comprehensive captions (for example, elucidating the symbols or variables employed) would serve to elevate clarity for the audience.

3. Standardize Terminology Consistency The manuscript occasionally oscillates between varying terminologies such as “programs,” “tasks,” or “jobs.” It is recommended to establish a standardized terminology throughout all sections to mitigate potential confusion.

4. Minor Linguistic Revisions Required The manuscript exhibits robust technical content; however, it contains instances of awkward phrasing (for example, “this heuristic presents itself as an augmentation…”). Such phrasing should be rectified.

5. Incorporate Further Discussion on Limitations. The discussion section would benefit from a brief acknowledgment of potential limitations, such as the presumption of uniform virtual machines or fixed job durations, and propose how these limitations might be addressed in subsequent research endeavors.

---

## Round 0.2 · accepted · Accept

Both reviewers are happy with this revision, and I am happy to accept this manuscript for publication in PeerJ Computer Science.

Reviewer 3 ·

Basic reporting

The author has addressed all of my comments from the first round. I recommend this article for publication.

Experimental design

The author has addressed all of my comments from the first round. I recommend this article for publication.

Validity of the findings

The author has addressed all of my comments from the first round. I recommend this article for publication.

Additional comments

The author has addressed all of my comments from the first round. I recommend this article for publication.

Reviewer 4 ·

Basic reporting

no comment

Experimental design

no comment

Validity of the findings

no comment

Additional comments

All reviewer comments have been carefully addressed, and corresponding revisions have been incorporated throughout the manuscript. The authors believe that the current version meets the journal’s standards in terms of clarity, technical rigor, and completeness, and the paper is now ready for publication.